# Exploring associations between social media addiction, social media fatigue, fear of missing out and sleep quality among university students: A cross-section study

Xinhong Zhu[‡]*, Taoyun Zheng[‡], Linlin Ding, Xiaona Zhang, Zhihan Li, Hao Jiang

School of Nursing, Hubei University of Chinese Medicine, Wuhan, China

‡ XZ and TZ are co-first authors on this work.
* zxh88@hbtcm.edu.cn

## Abstract

### Background

Social media use has been linked to poor sleep outcomes among university students in the cyber age, but the association between the negative consequences of social media use and sleep problems is not yet well understood. The present study investigated the relationships among social media usage, social media fatigue (SMF), fear of missing out (FoMO), social media addiction (SMA) and sleep quality in university students.

### Method

An online survey was administered to 2744 respondents that included the Pittsburgh Sleep Quality Index (PSQI); questionnaires evaluating FoMO, SMF, and SMA; and questions regarding sleep duration, social media use, health status, and demographic information.

### Result

A total of 19.9% of respondents suffered from sleep disturbance. A total of 15.6% of participants had sleep durations less than 5 h, and 21.6% of subjects had sleep durations longer than 9 h. Sleep quality was positively associated with SMF (OR = 1.387, 95% CI: 1.103~1.743), and SMA (OR = 1.415, 95% CI: 1.118~1.791). The relationship between FoMO and sleep disturbance was not significant. Compared to a sleep duration > 9 h, SMF increased the risk of shorter sleep durations [5–6 h sleep (OR = 2.226, 95% CI: 1.132~4.375), 6–7 h sleep (OR = 1.458, 95% CI: 1.061~2.002), and 7–8 h sleep (OR = 1.296, 95% CI: 1.007~1.670)]. FoMO and SMA did not significantly affect sleep duration. In addition, SMA (OR = 3.775, 95% CI: 3.141~4.537), FoMO (OR = 3.301, 95% CI: 2.753~3.958), and sleep disorders (OR = 1.284, 95% CI: 1.006~1.638) increased SMF.

### Conclusion

Upon experiencing negative consequences of social media use, such as SMF and SMA, university students were likely to experience sleep problems. Further research exploring the

**Data Availability Statement:** All relevant data are within the paper and its supporting information file.

**Funding:** This work was supported by the National Natural Science Foundation of China (No. 82003448), and Philosophy and Social Science Project of Hubei Provincial Department of Education (No. 21Q132). The funders had no role in study design, data collection and analysis, decision to publish, or preparation of the manuscript.

**Competing interests:** The authors have declared that no competing interests exist.

interventions that improve sleep and alleviate negative consequences of social media use should be conducted.

## 1. Introduction

Social media platforms have penetrated all aspects of university students' lives. According to the China Internet Network Information Center (CNNIC), the average weekly time spent online by Chinese netizens increased from 18.7 h in 2011 to 26.9 h in 2021 [1, 2]. Recent findings indicated that 99.6% of Chinese netizens used smartphones to access the internet, and 17.4% of Chinese internet users aged 20~29 years had internet dependence [1]. Multifunctional smartphone applications and consistent internet access are considered to explain the increased, and even excessive use of social media platforms [3]. For university students, social media is the main channel for communicating and sharing information, personal messages, opinions and ideas [4]. It was reported that approximately 74.5% of university students spent two to six hours on social media each day [5]. However, excessive social media use could have a series of negative consequences, such as problematic social media use [6], fear of missing out (FoMO) [7], social media fatigue (SMF) [8], social media addiction (SMA) [9], poor sleep quality [10, 11], poor mental health [10], and decreased academic performance [12].

## 2. Literature review and hypotheses

Individuals who experience FoMO may experience an increased level of negative emotions that may lead to impairments in social interactions [13]. There is a reciprocal relationship between FoMO and problematic social media use, which may lead individuals to experience SMF and SMA [14–16]. Constantly attending to social media may increase exhaustion levels and result in SMF [17]. SMF is defined as the subjective and negative feeling of tiredness and burnout due to social media use [17]. As time spent on social media platforms increased, an increasing number of university students experienced SMF and wanted to stop using social media [12, 18], which may contribute to a decrease in academic performance and use discontinuance [12, 19, 20]. SMA, the consequence of the compulsive use of social media platforms, manifests as behavioral addiction symptoms [21]. The pooled prevalence of SMA worldwide was found to be 5% (95% CI: 3%~7%) [22]. Accumulating evidence has shown that SMA is a growing problem among university students, and is related to self-esteem, life satisfaction, mental health and academic performance [21, 23–25]. Given that social media has become as a near-ubiquitous aspect of university students' lives, excessive social media use is becoming increasingly evident, and the negative consequences of excessive social media usage have attracted public concern.

Recently, increasing attention has been given to the relationship between sleep disorders and social media usage. University students frequently experience poor sleep [26, 27]. The prevalence of insomnia in university students worldwide has been found to be 18.5%, which is much higher than the prevalence in the general population (7.4%) [28]. The overall pooled prevalence rates of sleep disturbances and insomnia symptoms in Chinese university students are 25.7% (95% CI: 22.5~28.9%) and 23.6% (95% CI: 18.9~29.0%), respectively [29]. The mechanisms underlying the relationships between the use of social media and sleep problems are unclear, but a theoretical model of the relationship has been proposed [30] and suggests several possible mechanisms. According to this model, social media use may directly affect sleep quality by consuming excess time or interfere with sleep by altering psychological arousal

via observation of stimulating online content [30]. Relationships among delayed sleep onset, poor quality of sleep and increased time spent on the internet have been reported in previous studies [31, 32]. Frequent social media use predicted both poor mental health and poor sleep outcomes in youth [10]. Evidence showed that the impact of negative affect on sleep quality was mediated by FoMO and smartphone addiction in a sample of Chinese university students [33]. Additionally, insomnia partially mediated the significant associations of interpersonal stress and FoMO with mental health in college students [34]. Among individuals with internet addiction, the overall pooled odds ratio of experiencing sleep disturbance was 2.20 (95% CI: 1.77–2.74), and a significant reduction in sleep duration was observed [35]. Furthermore, a higher rate of insomnia was observed among Norwegian university students with higher levels of SMA [36]. In summary, FoMO and SMA are closely associated with sleep quality. Although the relationship between chronic fatigue and sleep quality was confirmed among university students [37], few studies on the relationship between SMF and sleep quality have been conducted among Chinese university students. Given that FoMO and SMA drive SMF [38], we hypothesized that SMF is related to sleep quality.

Although social media use is linked with poor sleep outcomes, few studies have focused on the relationships among FoMO, SMA, SMF and sleep quality. It is important to clarify the relationships among these four factors to determine the role of negative consequences of social media use on sleep quality and to develop interventions to improve university students' sleep quality in further studies. Thus, this study was aimed to investigate the relationships among social media use, FoMO, SMA, SMF, sleep disturbance and sleep duration in Chinese university students.

## 3. Methods

### 3.1 Participants and data collection

Using convenience sampling, 3015 participants from 4 universities and 4 vocational and technical colleges in Wuhan were selected among junior college students, and undergraduate students and above. The inclusion criteria for participants were voluntary participation in in the study and ability to cooperate with the study. Data were collected using structured questionnaires in February, 2021. Participants completed questionnaires online via Questionnaire Star. Data from 3015 respondents were included in the analysis; data from 271 respondents were excluded due to missing values.

### 3.2 Instruments

**Pittsburgh Sleep Quality Index (PSQI).**   The Chinese version of scale was developed by Liu et al. [39]. The scale contains 19 items in 7 subscales: subjective sleep quality, sleep latency, sleep duration, habitual sleep efficiency, sleep disturbances, use of sleep medication, and daytime dysfunction. The total score ranges from 0 to 21 points (range of 0 to 3 points for each subscale), with higher scores indicating worse sleep quality. The cutoff value for sleep disturbance is 7 [39]. In this study, the Cronbach's α value of the PSQI was 0.663.

**Sleep duration.**   We assessed sleep duration by asking respondents the following question: "How many hours did you usually sleep at night in the past month?" The responses for sleep duration were as follows: "< 5 h", "5–6 h", "6–7 h", "7–8 h", "8–9 h", and ">9 h".

**Social Media Fatigue (SMF).**   The scale was based on previous studies [8, 40, 41]. The tool contains 18 items on three subscales: anxiety (8 items), information value (4 items), escaping from social media (6 items). Responses to each item are rated on a five-point Likert scale ranging from 1 (strongly disagree) to 5 (strongly agree). The Cronbach's α values of the three dimensions in the present study were 0.918, 0.901 and 0.904, respectively. A high score

indicates a high level of SMF. The participants were divided into two groups using the median SMF score for statistical analysis: a high score group (≥50, n = 1374), and a low score group (<50, n = 1370).

**Social Media Addiction (SMA).** The scale was developed by Liu and Ma [23] and was originally used with college students in China. The tool contains 28 items rated on a 5-point Likert scale ranging from 1 (strongly disagree) to 5 (strongly agree). The Cronbach's α value of the SMA was 0.971 in this study. The participants were divided into two groups using the median SMA score for statistical analysis: a high score group (≥84, n = 1376), and a low score group (<84, n = 1368).

**Fear of Missing Out (FoMO).** The Chinese version of FoMO, which was adapted by Li et al following standardized international guidelines, was implemented to measure FoMO among university students in China [42, 43]. Each item is rated on a five-point Likert scale ranging from 1 (strongly disagree) to 5 (strongly agree). The Cronbach's α value for FoMO in the present study was 0.924. The participants were divided into two groups using the median FoMO score for statistical analysis: a high score group (≥32, n = 1398) and a low score group (<32, n = 1346).

**Social media use.** Internet use was ascertained with the following items: (1) "purposes of using social media" (to stay in touch with what my friends are doing, to research/find products to buy, to find funny or entertaining content, to learn, to stay up-to-date with news and current events, to play game, to share photos or videos with others, to initiate a topic, because a lot of my friends are on them, and others). (2) "number of social media accounts" (0~2, 3~4, 5~6, 7~8, 9~). (3) "time spent on social media per day" (0~2 h, 2~4 h, 4~6 h, 6~8 h, 8 h~). (4) "Do you spend more time on social media than real world?" (less, the same, slightly, much). (5) "browsing social media before bed" (strongly disagree, disagree, not agree, agree, strongly agree).

**Demographic questionnaire.** A demographic information sheet was used to acquire basic information, such as gender, age, residence, education level (junior college students, undergraduate students and above), single-child, parental marital status (married, single parent/stepparent, and others), and self-reported health status (good, fair, bad, chronic disease, history of serious illness, family history of serious illness).

### 3.3 Ethics statement

The University Research Ethics Committee of Hubei University of Chinese Medicine approved the study which was conducted in 2021 (2018-ICE-023). The study was carried out in accordance with the requisite ethical standards (e.g., the Helsinki declaration), and written informed consent was obtained from all participants. Prior to the collection of data, the purposes and procedures of this study were explained to the respondents. Participants were informed that they could withdraw from the study at any time. Data were collected only from those who voluntarily agreed to participate and provided written informed consent.

### 3.4 Data analysis

All analyses were performed using SPSS 20.0, and the significance threshold level of statistical tests was set at $p < 0.05$. Descriptive statistics of participants' demographic characteristics, internet use and SMF are described using number (n) and percentage (%), mean ± SD, or median and interquartile range. Logistic regression was used to identify significantly factors ($p < 0.05$) associated with sleep quality, sleep duration, social media use, SMF, SMA, FoMO, and demographic information. The binary regression results shown in Table 1 were obtained with the following steps: (1) In Model 1, FoMO, SMF and SMA were entered as independent variables, and sleep quality was entered as the dependent variable. (2) In Model 2, FoMO,

**Table 1. Factors associated with sleep quality among university students.**

| Variable | Model 1 | Model 2 | Model 3 |
|---|---|---|---|
| | OR (95%CI) | OR (95%CI) | OR (95%CI) |
| **SMF** | 1.480 (1.190~1.839) ** | 1.460 (1.169~1.824) * | 1.387 (1.103~1.743) * |
| **SMA** | 1.582 (1.270~1.972) ** | 1.391 (1.105~1.749) * | 1.415 (1.118~1.791) * |
| **FoMO** | 1.225 (0.987~1.521) | 1.132 (0.906~1.414) | 1.143 (0.910~1.435) |
| **Purposes** | | | |
| To stay up-to-date with news and current events (Yes) | | 0.801 (0.650~0.986) * | 0.834 (0.673~1.035) |
| To share photos or videos with others (Yes) | | 1.347 (1.064~1.706) * | 1.339 (1.050~1.708) * |
| **Do you spend more time on social media than real world?** | | | |
| Less | | Ref | Ref |
| The same | | 1.383 (1.022~1.872) * | 1.302 (0.956~1.773) |
| Slightly | | 1.571 (1.149~2.162) * | 1.520 (1.103~2.094) * |
| Much | | 2.007 (1.409~2.858) ** | 1.812 (1.259~2.607) * |
| Time spent on social media (h) | | | |
| 0~2 | | Ref | Ref |
| 2~4 | | 1.101 (0.801~1.513) | 1.120 (0.809~1.551) |
| 4~6 | | 1.301 (0.931~1.818) | 1.303 (0.922~1.841) |
| 6~8 | | 1.144 (0.753~1.738) | 1.175 (0.764~1.807) |
| 8~ | | 1.983 (1.290~3.048) * | 1.889 (1.208~2.953) * |
| **Education** | | | |
| Junior college students | | | Ref |
| Undergraduate students and above | | | 0.658 (0.533~0.812) ** |
| **Self-reported health status** | | | |
| Good | | | 0.617 (0.393~0.971) * |
| Fair | | | 1.795 (1.193~2.701) * |
| Bad | | | 3.474 (1.462~8.256) * |

*: $p < 0.05$

**: $p < 0.001$

Model 1was crude model; Model 2 was adjusted for social media use; Model 3 was adjusted for social media use and demographic information.

SMF, SMA and social media use were entered as independent variables. (3) In Model 3, demographic variables were entered as independent variables along with the variables in Model 2 to identify factors significantly affecting sleep quality. Then, variables were entered with $p < 0.25$ into a multivariate logistic regression analysis to identify predictors of sleep duration in univariate analysis. Multivariate logistic regression analyses were conducted with short sleep duration as the outcome variable ($> 9$ h as the reference category), and social media use, SMF, SMA, FoMO, and demographic information as the exposure variables (Table 2). In addition,

**Table 2. Factors associated with sleep duration among university students.**

| Variables | < 5 h [a] | 5–6 h [a] | 6–7 h [a] | 7–8 h [a] | 8–9 h [a] |
|---|---|---|---|---|---|
| | AOR (95%CI) | AOR (95%CI) | AOR (95%CI) | AOR (95%CI) | AOR (95%CI) |
| **SMF** | 0.984 (0.445~2.175) | 2.226 (1.132~4.375) * | 1.458 (1.061~2.002) * | 1.296 (1.007~1.670) * | 1.034 (0.814~1.314) |
| **SMA** | 1.848 (0.797~4.285) | 1.161 (0.588~2.294) | 1.110 (0.802~1.535) | 1.007 (0.778~1.305) | 1.046 (0.818~1.337) |
| **FoMO** | 1.007 (0.455~2.228) | 0.898 (0.473~1.703) | 1.020 (0.744~1.398) | 0.957 (0.744~1.232) | 0.994 (0.783~1.261) |

*: $p < 0.05$

[a]: $> 9$ h as reference category

**Table 3. Factors associated with SMF among university students.**

| Variable | Model 1 | Model 2 | Model 3 |
|---|---|---|---|
|  | OR (95%CI) | OR (95%CI) | OR (95%CI) |
| **SMA** | 3.872 (3.246~4.618) ** | 3.771 (3.142~4.526) ** | 3.775 (3.141~4.537) ** |
| **FoMO** | 3.276 (2.747~3.906) ** | 3.243 (2.708~3.882) ** | 3.301 (2.753~3.958) ** |
| **Sleep disturbance** | 1.363 (1.070~1.721) * | 1.340 (1.057~1.699) * | 1.284 (1.006~1.638) * |
| **Sleep duration** |  |  |  |
| < 5 h | 0.855 (0.400~1.828) | 0.870 (0.400~1.894) | 0.898 (0.410~1.966) |
| 5~6 h | 1.777 (0.904~3.495) | 1.972 (0.995~3.908) | 1.990 (0.995~3.982) |
| 6~7 h | 1.408 (1.024~1.935) * | 1.428 (1.035~1.970) * | 1.353 (0.975~1.877) |
| 7~8 h | 1.279 (1.003~1.632) * | 1.349 (1.052~1.730) * | 1.273 (0.986~1.643) |
| 8~9 h | 1.013 (0.801~ 1.281) | 1.049 (0.827~1.331) | 1.026 (0.806~1.306) |
| > 9 h | Ref | Ref | Ref |
| **Purposes of using social media** |  |  |  |
| Because a lot of my friends are on them |  | 1.372 (1.062~1.772) * | 1.432 (1.106~1.854) * |
| **Do you spend more time on social media than real world?** |  |  |  |
| Less |  | Ref | Ref |
| The same |  | 1.025 (0.806~1.302) | 1.023 (0.804~1.302) |
| Slightly |  | 1.240 (0.956~1.608) | 1.233 (0.949~1.601) |
| Much |  | 1.724 (1.252~2.373) * | 1.655 (1.198~2.286) * |
| **Education** |  |  |  |
| Junior college students |  |  | Ref |
| Undergraduate students and above |  |  | 1.310 (1.085~1.581) * |

*: $p < 0.05$

**: $p < 0.001$

Model 1was crude model; Model 2 was adjusted for Internet use; Model 3 was adjusted for Internet use and demographic information.

binary regression was fitted to access the factors affecting SMF after considering the effects of FoMO, SMA, sleep quality, sleep duration, social media use, and demographic information, as shown in the Table 3. The strength of the association was evaluated with odds ratios (ORs) and 95% confidence intervals (CIs). The models' goodness of fit was checked by using omnibus tests of model coefficients for overall fitness of the model and the Hosmer and Lemeshow test for fit of the model to the data.

## 4. Results

### 4.1 Participant characteristics

Participant characteristics are shown in S1 Table. The mean age of the respondents was 20.08 (SD = 2.37) years, with a range of 18~30 years old. Of the 2744 participants, 69.2% were female, 56.3% lived in rural areas and 69.4% were single-child.

### 4.2 Sleep quality

Using a PSQI score > 7 to identify sleep disturbance, among the 2744 participants, 2197 (80.1%) reported normal sleep quality and 547 (19.9%) reported sleep disturbance. As shown in Table 1, SMF (OR = 1.387, 95% CI: 1.103~1.743), and SMA (OR = 1.415, 95% CI: 1.118~1.791) were positively associated with sleep quality. When time spent on social media exceeded that spent in the real world, the sleep quality decreased. Self-reported health status

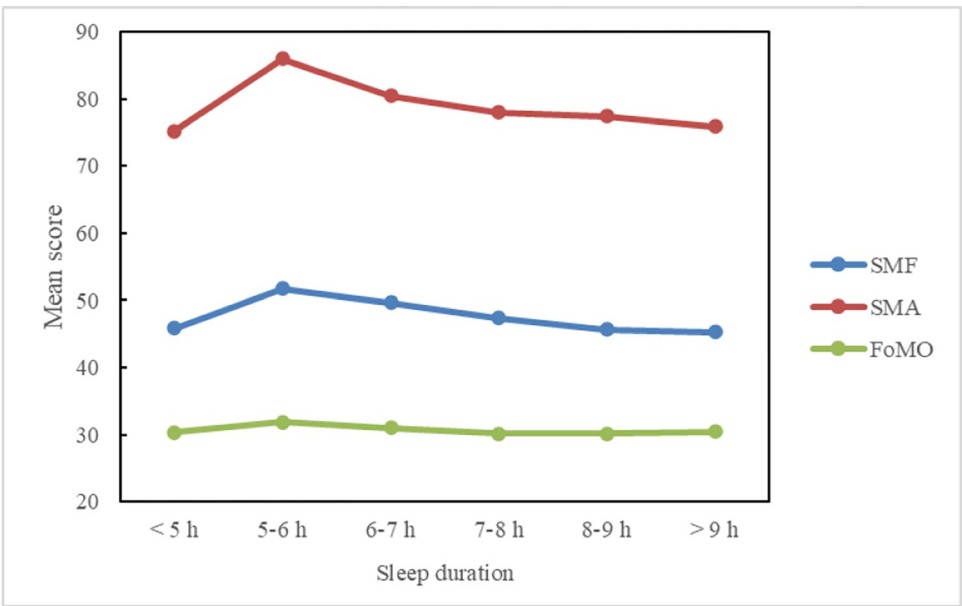

**Fig 1. Mean score of SMF, SMA and FoMO across sleep duration.**

[good (OR = 0.617, 95% CI: 0.393~0.971)], and education level [undergraduate students and above (OR = 0.658, 95%CI: 0.533~0.812)] negatively affected sleep quality.

### 4.3 Sleep duration

A total of 15.6% of participants reported having a short sleep duration (< 7 hours), and 21.6% of participants had sleep durations longer than 9 hours per day. As shown in Table 2, compared to a sleep duration > 9 h duration, SMF increased the risk of short sleep durations [5–6 h sleep (OR = 2.226, 95% CI: 1.132~4.375), and 7–8 h sleep (OR = 1.296, 95% CI: 1.007~1.670)]. FoMO and SMA did not significantly affect sleep duration. As shown in Fig 1, the mean scores of SMF, SMA and FoMO were highest at 5–6 h of sleep.

### 4.4 SMF

The median SMF score was 50 (25, 75). As shown in Table 3, SMA (OR = 3.775, 95% CI: 3.141~4.537), FoMO (OR = 3.301, 95% CI: 2.753~3.958), sleep quality (OR = 1.284, 95% CI: 1.006~1.638), purpose [because a lot of my friends are on them (OR = 1.432, 95% CI: 1.106~1.854)], time spent on social media exceeding that spent in the real world [much (OR = 1.655, 95% CI: 1.198~2.286)] and education level [undergraduate students and above (OR = 1.310, 95% CI: 1.085~1.581)] increased SMF. However, sleep duration did not affect SMF scores.

## 5. Discussion

Prolonged social media use for nonacademic purposes, FoMO, SMA, SMF, poor sleep quality and decreased social interactions in the real world were reported by the participants in this study. Social media use for nonacademic purposes decreased sleep quality. Social media use for nonacademic purposes can distract university students from learning, adversely affect their academic performance and social interactions, and lead to delayed bedtime [44]. Additionally, spending more time on social media than in the real world resulted in sleep disturbance,

which aligns with findings in a previous study [10]. Consistent with previous studies [45, 46], individuals with poor self-rated health and low education levels were more likely to experience poor sleep quality in this survey. This offers insights into potential targets for suggesting that minorities are particularly vulnerable to the effects of poor health and prolonged social media use on sleep quality.

In the context of social media, individuals may experience addiction and frustration induced by FoMO, which could contribute to sleep problems [16, 47]. However, FoMO did not affect sleep quality or duration in this study. It was reported that the sleep patterns of social media users may be disrupted when they are preoccupied and fear of missing out on any information on social media, which may not be well limited in time [48]. In addition, FoMO predicted shorter sleep durations as it drove late night social media use among the adolescents [49]. The inconsistent results may be explained by differences in the participants surveyed. Additionally, time spent on social media use at night, which is an important factor for sleep quality and duration, was not measured in this study. In this study, SMA positively influenced sleep quality, but was not associated with sleep duration. Furthermore, the relationship between SMA and sleep duration exhibited an inverted U curve among university students. Previous studies have confirmed that problematic social media use (operationalized as SMA) is associated with poor sleep quality [11, 33, 50], similar to our finding. The potential reasons that SMA may lead to sleep disturbance include portable smartphones and Wi-Fi access without spatial or temporal constraints that may be brought to bed. Using social media, especially before bed, may lead to difficulty falling asleep and delayed bedtime [51]. Delayed bedtime and late waking time may further lead to circadian rhythm desynchronization [31]. Additionally, light-emitting screens may suppress melatonin secretion [52]. However, a U-shaped curve between SMA and sleep duration was observed among Norwegian university students [36]. Among Korean school-age children, individuals who were at high risk for smartphone addiction were likely to have poor sleep quality and short sleep duration [53]. The different tools and cutoff values used to measure SMA and the different classifications of sleep duration may lead to inconsistent findings.

As university students heavily rely on various social media platforms to connect with others and use social media for nonacademic purposes, SMF was particularly evident. SMF is closely related to individuals' physical and mental health and may trigger unhealthy behaviors [54]. In this study, a reciprocally relationship between sleep quality and SMF was observed in university students. Individuals experiencing SMF exhibited sleep disturbance and short sleep duration. Likewise, short sleep duration and poor sleep quality exacerbated individuals' fatigue resulting from social media. Although few studies have focused on the relationship between SMF and sleep problems, there is a consensus that fatigue and sleep problems are associated [55, 56]. Evidence suggests that sleep disturbance may be an important contributor to fatigue in the context of students with depression and healthy employed people [57, 58]. In addition, cancer patients who reported being overly fatigued were 2.5 times more likely to have insomnia than others [59]. Moreover, FoMO was found to increase SMF, which is consistent with a previous study [15]. Furthermore, excessive social media use or SMA increases individual fatigue experience [60, 61]. Overall, individual-level factors (e. g. FoMO, decreased social interactions in the real world, addictive behaviors) play important roles in driving SMF. These findings also confirm the relationship between fatigue and sleep disturbance.

This study has some limitations. First, some of the study variables (e. g. SMA, FoMO, SMF, sleep duration and sleep quality) were reported using self-administered questionnaires. Therefore, our findings may be influenced by recall bias and social desirability. Second, we could not control for other factors, such as problematic social media use, nighttime social media use, weekend catch-up sleep, daytime naps, mobile phone use during social media outages,

nomophobia and insomnia. Thus, future studies should include these variables. Third, a cross-sectional design was used; thus, the causality of relationships among the study variables could not be determined. Furthermore, the convenience sampling technique used may limit the generalizability of the findings.

## 6. Conclusion

Poor sleep is a common phenomenon among university students. Although social media use is related to sleep quality, the relationship between the adverse effects of excessive social media use, especially fatigue from excessive social media use, and sleep quality has received less attention. In the present study, the results suggest that university students commonly experience sleep disturbance and insufficient sleep. Individuals with SMF and SMA are likely to experience poor sleep quality. Compared to a sleep duration > 9 hours, a short sleep duration was positively associated with SMF. In addition, SMA, FoMO, and sleep quality affected SMF. Further research exploring the relationships among nighttime social media use, weekend catch-up sleep, daytime naps, insomnia and SMF, and interventions to improve these conditions should be conducted.

## Supporting information

**S1 File.**
(XLSX)

**S1 Table. General characteristics and Internet use of the participants (n = 2744).**
(DOCX)

## Acknowledgments

We kindly thank Changjiang Polytechnic, Hubei University of Education, South-Central Minzu University, Wuhan Polytechnic, and 4 other universities and vocational colleges for their assistance in the enrollment and the assessment of participants.

## Author Contributions

**Conceptualization:** Xinhong Zhu.

**Investigation:** Taoyun Zheng, Linlin Ding, Xiaona Zhang, Zhihan Li, Hao Jiang.

**Supervision:** Xinhong Zhu.

**Validation:** Xinhong Zhu.

**Writing – original draft:** Xinhong Zhu.

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
