## [Decision Letter · Decision Letter 0]

6 Mar 2023

PONE-D-22-10183Exploring associations between social media addiction, social media fatigue, fear of missing out and sleep quality among university students: a cross-section studyPLOS ONE

Dear Dr. Zhu,

Thank you for submitting your manuscript to PLOS ONE. After careful consideration, we feel that it has merit but does not fully meet PLOS ONE’s publication criteria as it currently stands. Therefore, we invite you to submit a revised version of the manuscript that addresses the points raised during the review process.

We look forward to receiving your revised manuscript.

Kind regards,

Ugurcan Sayili, M.D.

Academic Editor

PLOS ONE

A clean copy of the edited manuscript (uploaded as the new *manuscript* file).

6. PLOS requires an ORCID iD for the corresponding author in Editorial Manager on papers submitted after December 6th, 2016. Please ensure that you have an ORCID iD and that it is validated in Editorial Manager. To do this, go to ‘Update my Information’ (in the upper left-hand corner of the main menu), and click on the Fetch/Validate link next to the ORCID field. This will take you to the ORCID site and allow you to create a new iD or authenticate a pre-existing iD in Editorial Manager. Please see the following video for instructions on linking an ORCID iD to your Editorial Manager account: https://www.youtube.com/watch?v=_xcclfuvtxQ.

Reviewers' comments:

Reviewer's Responses to Questions

**Comments to the Author**

1. Is the manuscript technically sound, and do the data support the conclusions?

Reviewer #1: Yes

Reviewer #2: Yes

Reviewer #3: Yes

2. Has the statistical analysis been performed appropriately and rigorously? 

Reviewer #1: Yes

Reviewer #2: Yes

Reviewer #3: Yes

3. Have the authors made all data underlying the findings in their manuscript fully available?

Reviewer #1: Yes

Reviewer #2: Yes

Reviewer #3: Yes

4. Is the manuscript presented in an intelligible fashion and written in standard English?

Reviewer #1: No

Reviewer #2: Yes

Reviewer #3: Yes

5. Review Comments to the Author

Reviewer #1: Comments on Exploring associations between social media addiction, social media fatigue, fear of missing out and sleep quality among university students: a cross-section study.

1.Paper summary

This paper mainly studied the associations between the phenomena of problems on social media and sleep quality based on Chinese university students.

2.comments

There are some problems which should be well solved before it is considered for publication.

First, relevant research background needs to be supplemented in Introduction. In this part, it is noted that the negative consequences of social media use, harmful effects of social media use, SMF, SMA and damaging relationship between social media use and sleep problems in paragraph 1 through 3 respectively. However, the summary or prospect of “little attention is on the combined impacts of adverse effects of excessive social media use” in line 80, consistency of the combined impacts didn’t be displayed in the following hypothesis and design.

Then, lots of silly mistakes are designed to get more attention.

① In Abstract. For poor sleep quality, OR of SMA, 1.409, conflicts with Table 1.

② In line 111, it is stated that model 3 had three groups, but only a high and a low group are described.

③ In line 118, participants were split into two subgroups using SMA, which included the number 2752, in contrast to the number of participants with the criteria of exclusion.

④ The description of the percentage of women and rural residents in line 164 conflicts with Table S1.

⑤ The statement in section 3.2 that "self-reported health status [not bad] (OR = 1.814)" conflicts with Table 1.

Reviewer #2: This research paper is well-structured and presents a thorough review of the literature. The analysis is clear and logical, and the conclusions are well-supported. The paper is well-written and easy to understand.

The findings of this study suggest that SMF, FoMO, and SMA are all significantly associated with sleep quality among university students. Specifically, higher levels of SMF, FoMO, and SMA were associated with decreased sleep quality. These results suggest that university students with higher levels of social media fatigue, fear of missing out, and social media addiction may be more likely to experience poorer sleep outcomes. These findings could have important implications for interventions that seek to improve the sleep outcomes of university students. It may be beneficial for such interventions to target these specific factors, such as SMF, FoMO, and SMA, to help improve sleep quality. Additionally, healthcare providers should be aware of the potential impact of these factors on university student sleep quality and consider addressing them in clinical care.

The research is relevant but the main problem is not up-to-date, and the sources need update and discussion. Mobile phone use during social media outage, nomophobia, and insomnia literature not been discussed which are all FOMO.

Reviewer #3: 1.It is appropriate for researchers to specify p values in a single format (with a lowercase p) throughout the article.

2.In the methodology section, the sleep duration categories should be corrected as “<7 h”, “7-8 h”, and “>8 h”.

3.Logistic regression is used to identify predictors of dependent variables, if more than one independent predictor variable is evaluated, the p value in univariate analysis p<0.25* is tested using multivariate logistic regression analysis of clinically significant variables. A statement should be made that the variables observed in the text and in the table were tested using multivariate logistic regression analysis (* David Hosmer, Stanley Lemeshow, Rodney Sturdivant - Applied Logistic Regression-Wiley (2013), page 91).

6. PLOS authors have the option to publish the peer review history of their article (what does this mean?). If published, this will include your full peer review and any attached files.

Reviewer #1: No

Reviewer #2: No

Reviewer #3: No

---

## [Author Response · Author response to Decision Letter 0]

6 Apr 2023

Dear. Ugurcan Sayili, M.D:

On behalf of my co-authors, I thank you very much for giving us an opportunity to revise our manuscript, we also deeply appreciate the editor and reviewers for their positive and constructive comments and suggestions on our manuscript entitled “Exploring associations between social media addiction, social media fatigue, fear of missing out and sleep quality among university students: a cross-section study” (ID: PONE-D-22-10183). Those comments are all valuable and very helpful for revising and improving our paper, as well as providing important guidance for our research.

We have studied editor and reviewers’ comments carefully and have made corresponding revisions that we hope to meet with qualifications for publication. We have tried our best to revise our manuscript according to the comments. Attached please find the revised version, which we would like to submit for your kind consideration. The major changes and revisions have been highlighted in red, which can be turned into black by rejecting all the changes. We hope that these revisions are satisfactory for your serious consideration to publish this manuscript in Plos one. We would like to express our appreciation again to you and reviewers for the comments on our paper. 

Looking forward to hearing back from you.

Correspondence and phone calls about the paper should be directed to Xinhong Zhu at the following address, phone, and e-mail address:

Dr. Xinhong Zhu

School of Nursing, Hubei University of Chinese Medicine, 430061 16# Huangjiahu West Road, Wuhan, China, E-mail: zxh88@hbtcm.edu.cn

Thanks for your attention to our paper.

We are very grateful to the reviewers and editor for their comments and suggestions for revisions. The manuscript has been substantially improved according to the suggestions of reviewers and editor. The main corrections in the paper and the responses to the reviewer’s comments are addressed below.

Journal Requirements:

Response: We really thank you for your careful review on our manuscript. We have downloaded MANUSCRIPT BODY FORMATTING GUIDELINES and TITLE, AUTHOR, AFFILIATIONS FORMATTING GUIDELINES and revised manuscript to meet PLOS ONE's style requirements.

A clean copy of the edited manuscript (uploaded as the new *manuscript* file).

Response: Thank you very much for your serious review and kind reminding. We invited Pro. Fen Yan to provide language editing.

Fen Yang: Nurse educator, Health Care Management, School of Nursing, Hubei University of Chinese Medicine,

So far, she has published 28 articles. Recent articles as corresponding author are as follows

1. Li, C., Wu, M., Qiao, G., Gao, X., Hu, T., Zhao, X., Zhu, X., & Yang, F. (2023). Effectiveness of Continuity of Care in Reducing Depression Symptoms in Elderly: A Systematic Review and Meta‐analysis. International Journal of Geriatric Psychiatry, e5894.

2. Hu, T., Zhao, X., Wu, M., Li, Z., Luo, L., Yang, C., & Yang, F. (2022). Prevalence of depression in older adults: A systematic review and meta-analysis. Psychiatry research, 114511.

3. Zhao, X., Hu, T., Qiao, G., Li, C., Wu, M., Yang, F., & Zhou, J. (2022). Psychometric properties of the smartphone distraction scale in Chinese college students: validity, reliability and influencing factors. Frontiers in psychiatry, 13.

4. Li, C., Yang, F., Yang, B.X. et al. Experiences and challenges faced by community mental health workers when providing care to people with mental illness: a qualitative study. BMC Psychiatry 22, 623 (2022).

Response: Thank you very much for your serious review and kind reminding. We have modified grant numbers of funds, which was consist with “Financial Disclosure” in the manuscript.

Response: We sincerely thank for your suggestion. We modified the part of “Data Availability Statement”.

All relevant data are within the manuscript and its Supporting information file.

Response: We sincerely thank for your suggestion. All relevant data are within the manuscript and its Supporting information file. 

6. PLOS requires an ORCID iD for the corresponding author in Editorial Manager on papers submitted after December 6th, 2016. Please ensure that you have an ORCID iD and that it is validated in Editorial Manager. To do this, go to ‘Update my Information’ (in the upper left-hand corner of the main menu), and click on the Fetch/Validate link next to the ORCID field. This will take you to the ORCID site and allow you to create a new iD or authenticate a pre-existing iD in Editorial Manager. Please see the following video for instructions on linking an ORCID iD to your Editorial Manager account: https://www.youtube.com/watch?v=_xcclfuvtxQ.

Response: Great thanks to you for your careful review and valuable suggestion. ORCID iD of corresponding author is shown as followed.

Xinhong Zhu: https://orcid.org/0000-0001-7356-7401

Reviewers' comments:

Reviewer #1:

Comments on Exploring associations between social media addiction, social media fatigue, fear of missing out and sleep quality among university students: a cross-section study.

1. Paper summary

This paper mainly studied the associations between the phenomena of problems on social media and sleep quality based on Chinese university students.

2. comments

Comment 1: There are some problems which should be well solved before it is considered for publication.

First, relevant research background needs to be supplemented in Introduction. In this part, it is noted that the negative consequences of social media use, harmful effects of social media use, SMF, SMA and damaging relationship between social media use and sleep problems in paragraph 1 through 3 respectively. However, the summary or prospect of “little attention is on the combined impacts of adverse effects of excessive social media use” in line 80, consistency of the combined impacts didn’t be displayed in the following hypothesis and design.

Response: We sincerely thank Reviewer 1 for your kind and professional comments of our paper, which not only encourage us to improve our present manuscript, but also provide some useful ideas for our future studies. Accordingly, we have studied comments carefully. Point-by-point replies to Reviewer #1 are listed below. We modified the paragraph 3, and rewrote the relationship of social media use (e.g. FoMO, SMA, and SMF) and sleep quality, and added 9 new references into the text. Meanwhile, the hypothesis of this study was revised.

 Paragraph 3-4:

 Recently, studies concerning social media use have stressed growing problems among social media in terms of sleep disorders. The relationships of delayed sleep onset, bad quality of sleep and increased time spent on internet were reported in previous studies [27, 28]. Frequent social media use could predict both poor mental health and sleep outcomes in youth [10]. Evidence showed that the impact of negative affect on sleep quality was mediated by FoMO and smartphone addiction in the sample of Chinese university students [29]. Additionally, higher level of FoMO was associated with higher level of insomnia, which in turn was associated with poorer mental health in the college students [30]. When individuals were addicted to the Internet, the overall pooled odds ratio of experiencing sleep disturbance was 2.20 (95% CI: 1.77–2.74), and a significant reduce in sleep duration was observed [31]. Furthermore, there was higher rates of insomnia among those with higher levels of SMA among Norwegian university students [32]. Although the relationship between chronic fatigue and sleep quality was confirmed among university students [33], few surveys on association of SMF and sleep quality were conducted among Chines university students. Given FoMO and SMA acting as SMF drivers [34], we hypothesized that SMF was related to sleep quality.

University students frequently suffer from poor sleep [35, 36]. The prevalence of insomnia in university students worldwide has been found to be 18.5%, much higher than the 7.4% rate in the general population [37]. The overall pooled prevalence of sleep disturbances and those suffering from insomnia symptoms in Chinese university students are 25.7% (95% CI: 22.5 ~28.9%) and 23.6% (95% CI: 18.9~29.0%), respectively [38]. Considering that social media use has been linked to poor sleep outcomes, we hypothesized that there were relationships of sleep quality with FoMO, SMF, and SMA. Thus, this study was conducted among Chinese university students, and correlated variables were investigated in detail, such as social media use, FoMO, SMA, SMF, sleep disturbances and sleep duration. 

① In Abstract. For poor sleep quality, OR of SMA, 1.409, conflicts with Table 1.

② In line 111, it is stated that model 3 had three groups, but only a high and a low group are described.

③ In line 118, participants were split into two subgroups using SMA, which included the number 2752, in contrast to the number of participants with the criteria of exclusion.

④ The description of the percentage of women and rural residents in line 164 conflicts with Table S1.

⑤ The statement in section 3.2 that "self-reported health status [not bad] (OR = 1.814)" conflicts with Table 1.

Response: We sincerely thank you for your kind and professional comments on our manuscript. We apologize for our carelessness. In this study, because grouping of respondents based upon the midpoints of the score ranges produces very unequal numbers in the two categories, we have modified the cut-off of SMA, SMF and FoMO, and used the median values as the cut-off. The median scores of SMA, SMF and FoMO were 84, 50 and 32, respectively. 

Meanwhile, we have modified our mistakes as followed:

① In the abstract, for poor sleep quality, the OR of SMA is 1.415, which is consistent with Table 1.

② The participants were divided into two groups using the median SMF score for statistical analysis: a high score group (≥50, n = 1374), and a low score group (<50, n = 1370).

③ The participants were divided into two groups using the median SMA score for statistical analysis: a high score group (≥84, n = 1376), and a low score group (<84, n = 1368). 

④ Of the 2744 participants, 69.2% were female, 56.3% lived in rural and 69.4% were single-child, which are consistent with data shown in Table S1.

⑤ Self-reported health status [fair (OR = 1.795, 95% CI: 1.193~2.701), bad (OR = 3.474, 95%CI: 1.462~8.256)] affected sleep quality. 

Thank you again for your comments on our manuscripts. 

Reviewer #2: This research paper is well-structured and presents a thorough review of the literature. The analysis is clear and logical, and the conclusions are well-supported. The paper is well-written and easy to understand.

The findings of this study suggest that SMF, FoMO, and SMA are all significantly associated with sleep quality among university students. Specifically, higher levels of SMF, FoMO, and SMA were associated with decreased sleep quality. These results suggest that university students with higher levels of social media fatigue, fear of missing out, and social media addiction may be more likely to experience poorer sleep outcomes. These findings could have important implications for interventions that seek to improve the sleep outcomes of university students. It may be beneficial for such interventions to target these specific factors, such as SMF, FoMO, and SMA, to help improve sleep quality. Additionally, healthcare providers should be aware of the potential impact of these factors on university student sleep quality and consider addressing them in clinical care.

Comment 1: The research is relevant but the main problem is not up-to-date, and the sources need update and discussion. Mobile phone use during social media outage, nomophobia, and insomnia literature not been discussed which are all FOMO.

Response: We appreciate Reviewer #2 for warm work and positive suggestions earnestly, which are very helpful for improving our paper and future research. In the paragraph 3, we modified the hypothesis and added evidence for supporting the relationship of FoMO and sleep problems as followed. 

Recently, studies concerning social media use have stressed growing problems among social media in terms of sleep disorders. The relationships of delayed sleep onset, bad quality of sleep and increased time spent on internet were reported in previous studies [26, 27]. Frequent social media use could predict both poor mental health and sleep outcomes in youth [10]. Evidence showed that the impact of negative affect on sleep quality was mediated by FoMO and smartphone addiction in the sample of Chinese university students [28]. Additionally, higher level of FoMO was associated with higher level of insomnia, which in turn was associated with poorer mental health in the college students [29]. When individuals were addicted to the Internet, the overall pooled odds ratio of experiencing sleep disturbance was 2.20 (95% CI: 1.77–2.74), and a significant reduce in sleep duration was observed [30]. Furthermore, there was higher rates of insomnia in those with higher levels of SMA among Norwegian university students [31]. Although the relationship between chronic fatigue and sleep quality was confirmed among university students [32], few surveys on association of SMF and sleep quality were conducted among Chinese university students. Given FoMO and SMA acting as SMF drivers [33], we hypothesized that SMF was related to sleep quality. 

Meanwhile, we modified the part of limitation as followed:

 Second, the study could not control for other factors, such as problematic social media use, nighttime social media use, weekend catch-up sleep, daytime nap, mobile phone use during social media outage, nomophobia and insomnia. Thus, future studies should include these variables. 

Reviewer #3: 

Comment 1: 1. It is appropriate for researchers to specify p values in a single format (with a lowercase p) throughout the article.

Response: Thank you very much for your serious review and kind reminding. We revised p values in a single format (with a lowercase p) in the part of methods and results.

Comment 2: 2. In the methodology section, the sleep duration categories should be corrected as “<7 h”, “7-8 h”, and “>8 h”.

Response: Great thanks to you for your careful review and valuable suggestion. In fact, if sleep duration categories were be corrected as “<7 h”, “7-8 h”, and “>8 h” in this study, the associations of sleep duration, SMA, SMF and FoMO were not observed. 

Therefore, in order to explore relationship of sleep duration, SMA, SMF and FoMO, sleep duration was divided into six categories: “< 5 h”, “5-6 h”, “6-7 h”, “7-8 h”, “8-9 h”, and “>9 h” in this study. As shown in Table 2, compared to > 9 h duration, SMF was associated with an increased risk of 5-6 h sleep (OR = 2.226, 95%CI: 1.132~4.375), while the risk of 7-8 h sleep was OR = 1.296 (95%CI: 1.007~1.670). FoMO and SMA did not significantly affect sleep duration. As shown in Fig 1, the mean scores of SMF, SMA and FoMO were highest at 5-6 h sleep.

Comment 3: 3. Logistic regression is used to identify predictors of dependent variables, if more than one independent predictor variable is evaluated, the p value in univariate analysis p<0.25* is tested using multivariate logistic regression analysis of clinically significant variables. A statement should be made that the variables observed in the text and in the table were tested using multivariate logistic regression analysis (* David Hosmer, Stanley Lemeshow, Rodney Sturdivant - Applied Logistic Regression-Wiley (2013), page 91).

Response: We deeply appreciate for your careful review on our manuscript. We modified the part of data analysis as followed:

 Then, variables were entered with p < 0.25 into a multivariate logistic regression analysis to identify predictors of sleep duration in univariate analysis. Multivariate logistic regression analyses were conducted with short sleep duration as the outcome variable (> 9 h as the reference category), and social media use, SMF, SMA, FoMO, and demographic information as the exposure variables (Table 2).

---

## [Decision Letter · Decision Letter 1]

3 May 2023

PONE-D-22-10183R1Exploring associations between social media addiction, social media fatigue, fear of missing out and sleep quality among university students: a cross-section studyPLOS ONE

Dear Dr. Zhu,

Thank you for submitting your manuscript to PLOS ONE. After careful consideration, we feel that it has merit but does not fully meet PLOS ONE’s publication criteria as it currently stands. Therefore, we invite you to submit a revised version of the manuscript that addresses the points raised during the review process.

We look forward to receiving your revised manuscript.

Kind regards,

Ahsan Ali

Academic Editor

PLOS ONE

Additional Editor Comments:

I am grateful to the authors for their efforts in addressing the comments of the reviewers. However, I would like to suggest a few points before the paper moves on to the next step:

1. I recommend that the authors revise the abstract. In the "Background" section, the authors mention that "Social media use has been linked to poor sleep outcomes in the cyber age among university students, but the mechanisms underlying this association are not yet well understood." It would be appropriate to specify which mechanisms the authors studied in this article rather than making a general statement about the relationships. Additionally, a clear description of the results is needed in the abstract. Please avoid ambiguous statements such as "Poor sleep quality was significantly with SMF." Please write clearly.

2. Please clarify how this study contributes to the existing literature and what it offers for practice in the last paragraph of the introduction section. Please align this with the discussion section as well.

3. Furthermore, I suggest that the authors include another heading after the introduction to provide an explanation of the relationship they have studied in this research and why these relationships are proposed. Also explain the theoretical foundation and practical evidence of the proposed relationships.

4. Lastly, I appreciate the language improvement, but there are still a number of grammatical issues along with unclear expressions. Therefore, I recommend that the authors use the services of a professional copy-editor to help with this issue.

Reviewers' comments:

Reviewer's Responses to Questions

**Comments to the Author**

1. If the authors have adequately addressed your comments raised in a previous round of review and you feel that this manuscript is now acceptable for publication, you may indicate that here to bypass the “Comments to the Author” section, enter your conflict of interest statement in the “Confidential to Editor” section, and submit your "Accept" recommendation.

Reviewer #2: All comments have been addressed

Reviewer #3: All comments have been addressed

2. Is the manuscript technically sound, and do the data support the conclusions?

Reviewer #2: Yes

Reviewer #3: Yes

3. Has the statistical analysis been performed appropriately and rigorously? 

Reviewer #2: Yes

Reviewer #3: Yes

4. Have the authors made all data underlying the findings in their manuscript fully available?

Reviewer #2: (No Response)

Reviewer #3: Yes

5. Is the manuscript presented in an intelligible fashion and written in standard English?

Reviewer #2: Yes

Reviewer #3: Yes

6. Review Comments to the Author

Reviewer #2: I am pleased to confirm that all comments have been thoroughly addressed and I am fully satisfied with the feedback provided to the authors.

Reviewer #3: The researchers made the desired adjustments. Acceptance of the article is appropriate and congratulations to the authors.

7. PLOS authors have the option to publish the peer review history of their article (what does this mean?). If published, this will include your full peer review and any attached files.

Reviewer #2: No

Reviewer #3: **Yes: **Aydin, Sumeyye Nur

---

## [Author Response · Author response to Decision Letter 1]

10 May 2023

Dear. Ahsan Ali

 On behalf of my co-authors, I thank you very much for giving us an opportunity to revise our manuscript, we also deeply appreciate the editor for your positive and constructive comments and suggestions on our manuscript entitled “Exploring associations between social media addiction, social media fatigue, fear of missing out and sleep quality among university students: a cross-section study” (ID: PONE-D-22-10183. R2). Those comments are all valuable and very helpful for revising and improving our paper, as well as providing important guidance for our research.

We have studied editor’s comments carefully and have made corresponding revisions that we hope to meet with qualifications for publication. We have tried our best to revise our manuscript according to the comments. Attached please find the revised version, which we would like to submit for your kind consideration. The major changes and revisions have been highlighted in red, which can be turned into black by rejecting all the changes. We hope that these revisions are satisfactory for your serious consideration to publish this manuscript in Plos One. We would like to express our appreciation again to you and reviewers for the comments on our paper. 

Looking forward to hearing back from you.

Correspondence and phone calls about the paper should be directed to Xinhong Zhu at the following address, phone, and e-mail address:

Dr. Xinhong Zhu

School of Nursing, Hubei University of Chinese Medicine, 430061 16# Huangjiahu West Road, Wuhan, China, E-mail: zxh88@hbtcm.edu.cn

Thanks for your attention to our paper.

We are very grateful to the editor for your comments and suggestions for revisions. The manuscript has been substantially improved according to the suggestions of editor. The main corrections in the paper and the responses to editor’s comments are addressed below.

I am grateful to the authors for their efforts in addressing the comments of the reviewers. However, I would like to suggest a few points before the paper moves on to the next step:

Comment 1. I recommend that the authors revise the abstract. In the "Background" section, the authors mention that "Social media use has been linked to poor sleep outcomes in the cyber age among university students, but the mechanisms underlying this association are not yet well understood." It would be appropriate to specify which mechanisms the authors studied in this article rather than making a general statement about the relationships. Additionally, a clear description of the results is needed in the abstract. Please avoid ambiguous statements such as "Poor sleep quality was significantly with SMF." Please write clearly.

Response: We sincerely thank your kind and professional comments of our paper, which not only encourage us to improve our present manuscript, but also provide some useful ideas for our future studies. Accordingly, we have studied comments carefully. Point-by-point replies to editor are listed below. We had modified the part of abstract.

Background: Social media use has been linked to poor sleep outcomes in the cyber age among university students, but the association between negative consequences of social media use and sleep problems are not yet well understood. The present study investigated the relationships of social media usage, social media fatigue (SMF), fear of missing out (FoMO), social media addiction (SMA) and sleep quality among university students.

Method: Online survey was conducted with 2744 respondents, who completed questionnaires including the Pittsburgh Sleep Quality Index (PSQI), FoMO, SMF, SMA and questions regarding sleep duration, social media use, health status, and demographic information.

Result: 19.9% of respondents suffered from sleep disturbance. 15.6% of participants slept less than 5 h, and 21.6% of subjects slept more than 9 h. Sleep quality was positively associated with SMF (OR = 1.387, 95% CI: 1.103~1.743), and SMA (OR = 1.415, 95% CI: 1.118~1.791). The relationship between FoMO and sleep disturbance was not significant. Compared to > 9 h duration, SMF increased risk of 5-6 h sleep (OR = 2.226, 95%CI: 1.132~4.375), 6-7 h sleep (OR = 1.458, 95%CI: 1.061~2.002), and 7-8 h sleep (OR =1.296, 95%CI: 1.007~1.670). FoMO and SMA did not significantly affect sleep duration. In addition, SMA (OR = 3.775, 95%CI: 3.141~4.537), FoMO (OR = 3.301, 95%CI: 2.753~3.958), and sleep disorder (OR = 1.284, 95%CI: 1.006~1.638) positively affected SMF level.

Conclusion: On experiencing negative consequences of social media use, such as SMF and SMA, university students are likely to have sleep problems. Further research exploring the interventions for improving poor sleep and negative outcomes of social media use should be conducted. 

Comment 2. Please clarify how this study contributes to the existing literature and what it offers for practice in the last paragraph of the introduction section. Please align this with the discussion section as well.

Response: Thank you very much for your serious review and kind reminding. We had modified the last paragraph of the introduction section.

 Although social media use has been linked to poor sleep outcomes, little research has focused on the relationship of FoMO, SMA, SMF and sleep quality. It is important to clarify the relationships of these four factors in order to detect the role of negative outcomes of social media use on sleep quality, and develop interventions to improve university students’ sleep quality for further study. Thus, this study was aimed to investigated the relationship between social media use, FoMO, SMA, SMF, sleep disturbances and sleep duration among Chinese university students.

Comment 3. Furthermore, I suggest that the authors include another heading after the introduction to provide an explanation of the relationship they have studied in this research and why these relationships are proposed. Also explain the theoretical foundation and practical evidence of the proposed relationships.

Response: Great thanks to you for your careful review and valuable suggestion. We had added another heading “Literature review and hypotheses” after the introduction and revised paragraph 3 and 4 as followed.

 Recently, growing attention was paid on relationship of sleep disorders and social media usage. University students frequently suffer from poor sleep [26, 27]. The prevalence of insomnia in university students worldwide has been found to be 18.5%, much higher than the 7.4% rate in the general population [28]. The overall pooled prevalence of sleep disturbances and those suffering from insomnia symptoms in Chinese university students are 25.7% (95% CI: 22.5 ~28.9%) and 23.6% (95% CI: 18.9~29.0%), respectively [29]. The mechanisms behind the relationships between use of social media and sleep problems are not well established, but a theoretical model of the relationship has been proposed [30], suggesting several possible mechanisms. According to this model, social media use directly affect sleep quality by excess time consuming, or it may interfere with sleep by psychological arousal through the stimulating online content [30]. The relationships of delayed sleep onset, bad quality of sleep and increased time spent on internet were reported in previous studies [31, 32]. Frequent social media use could predict both poor mental health and bad sleep outcomes in youth [10]. Evidence showed that the impact of negative affect on sleep quality was mediated by FoMO and smartphone addiction in the sample of Chinese university students [33]. Additionally, insomnia partially mediated significant associations of interpersonal stress and FoMO with mental health in the college students [34]. When individuals were addicted to the Internet, the overall pooled odds ratio of experiencing sleep disturbance was 2.20 (95% CI: 1.77–2.74), and a significant reduce in sleep duration was observed [35]. Furthermore, there was higher rate of insomnia in those with higher level of SMA among Norwegian university students [36]. In summary, FoMO and SMA are closely with sleep quality. Although the relationship between chronic fatigue and sleep quality was confirmed among university students [37], few surveys on association of SMF and sleep quality were conducted among Chinese university students. Given FoMO and SMA acting as SMF drivers [38], we hypothesized that SMF was related to sleep quality. 

Although social media use is linked with poor sleep outcomes, little research has focused on the relationship of FoMO, SMA, SMF and sleep quality. It is important to clarify the relationships of these four factors in order to detect the role of negative outcomes of social media use on sleep quality, and develop interventions to improve university students’ sleep quality for further study. Thus, this study was aimed to investigate the relationship between social media use, FoMO, SMA, SMF, sleep disturbance and sleep duration among Chinese university students.

Comment 4. Lastly, I appreciate the language improvement, but there are still a number of grammatical issues along with unclear expressions. Therefore, I recommend that the authors use the services of a professional copy-editor to help with this issue.

Response: We deeply appreciate for your careful review on our manuscript. We invited professor to improve the grammatical issues.

---

## [Decision Letter · Decision Letter 2]

29 Aug 2023

PONE-D-22-10183R2Exploring associations between social media addiction, social media fatigue, fear of missing out and sleep quality among university students: a cross-section studyPLOS ONE

Dear Dr. Zhu,

Thank you for submitting your manuscript to PLOS ONE. After careful consideration, we feel that it has merit but does not fully meet PLOS ONE’s publication criteria as it currently stands. Therefore, we invite you to submit a revised version of the manuscript that addresses the points raised during the review process.

We look forward to receiving your revised manuscript.

Kind regards,

Ahsan Ali

Academic Editor

PLOS ONE

Journal Requirements:

Additional Editor Comments:

I appreciate your submission of the manuscript titled "Exploring associations between social media addiction, social media fatigue, fear of missing out, and sleep quality among university students: a cross-sectional study" to Plos One. After a rigorous evaluation conducted by multiple experts in the field, I have made a decision regarding the publication of your manuscript.

While your work addresses an interesting and significant topic, and the reviewers have noted substantial improvements in the paper, there remains a concern regarding language and grammar issues. To ensure the highest quality of publication, I strongly recommend that the authors consider seeking professional copy editing services to address these language and grammar concerns. Consequently, I invite you to revise and resubmit the paper after addressing these concerns.

Reviewers' comments:

Reviewer's Responses to Questions

**Comments to the Author**

1. If the authors have adequately addressed your comments raised in a previous round of review and you feel that this manuscript is now acceptable for publication, you may indicate that here to bypass the “Comments to the Author” section, enter your conflict of interest statement in the “Confidential to Editor” section, and submit your "Accept" recommendation.

Reviewer #4: All comments have been addressed

2. Is the manuscript technically sound, and do the data support the conclusions?

Reviewer #4: (No Response)

3. Has the statistical analysis been performed appropriately and rigorously? 

Reviewer #4: Yes

4. Have the authors made all data underlying the findings in their manuscript fully available?

Reviewer #4: Yes

5. Is the manuscript presented in an intelligible fashion and written in standard English?

Reviewer #4: (No Response)

6. Review Comments to the Author

Reviewer #4: I would like to congratulate the authors for highlighting the prevailing issues regarding over use of social media among university students, and extending the existing body of knowledge.

7. PLOS authors have the option to publish the peer review history of their article (what does this mean?). If published, this will include your full peer review and any attached files.

Reviewer #4: **Yes: **Tahir Ashfaq

---

## [Author Response · Author response to Decision Letter 2]

1 Sep 2023

Dear. Ahsan Ali:

On behalf of my co-authors, I thank you very much for giving us an opportunity to revise our manuscript, we also deeply appreciate the editor for your positive and constructive comments and suggestions on our manuscript entitled “Exploring associations between social media addiction, social media fatigue, fear of missing out and sleep quality among university students: a cross-section study” (ID: PONE-D-22-10183.R3). Those comments are all valuable and very helpful for revising and improving our paper, as well as providing important guidance for our research.

We have studied editor’s comments carefully and have made corresponding revisions that we hope to meet with qualifications for publication. We have tried our best to revise our manuscript according to the comments. Attached please find the revised version, which we would like to submit for your kind consideration. The major changes and revisions have been highlighted in red, which can be turned into black by rejecting all the changes. We hope that these revisions are satisfactory for your serious consideration to publish this manuscript in the PLOS ONE. We would like to express our appreciation again to you and reviewer for the comments on our paper. 

Looking forward to hearing back from you.

Correspondence and phone calls about the paper should be directed to Xinhong Zhu at the following address, phone, and e-mail address:

Dr. Xinhong Zhu

We are very grateful to your comments and suggestions for revisions. The manuscript has been substantially improved according to the suggestions of the editor. The main corrections in the paper and the responses to the editor’s comments are addressed below.

Comment 1

I appreciate your submission of the manuscript titled "Exploring associations between social media addiction, social media fatigue, fear of missing out, and sleep quality among university students: a cross-sectional study" to Plos One. After a rigorous evaluation conducted by multiple experts in the field, I have made a decision regarding the publication of your manuscript. 

While your work addresses an interesting and significant topic, and the reviewers have noted substantial improvements in the paper, there remains a concern regarding language and grammar issues. To ensure the highest quality of publication, I strongly recommend that the authors consider seeking professional copy editing services to address these language and grammar concerns. Consequently, I invite you to revise and resubmit the paper after addressing these concerns.

Response: We deeply appreciate your comments. We have paid for a professional English editing service in AJE, and improved the quality of written English, including spelling mistakes, sentence structure and verb tense.

---

## [Editor Report · Decision Letter 3]

21 Sep 2023

Exploring associations between social media addiction, social media fatigue, fear of missing out and sleep quality among university students: a cross-section study

PONE-D-22-10183R3

Dear Dr. Zhu,

We’re pleased to inform you that your manuscript has been judged scientifically suitable for publication and will be formally accepted for publication once it meets all outstanding technical requirements.

Kind regards,

Ahsan Ali

Academic Editor

PLOS ONE

Additional Editor Comments (optional):

I am thankful to you for submitting the article titled "Exploring Associations Between Social Media Addiction, Social Media Fatigue, Fear of Missing Out, and Sleep Quality Among University Students: A Cross-Sectional Study" to PLOS ONE. I have reviewed the article along with the insightful comments provided by the three independent reviewers. Based on my reviewer recommendations and my assessment, I am pleased to recommend the acceptance of this paper for publication.
---

## [Editor Report · Acceptance letter]

26 Sep 2023

PONE-D-22-10183R3 

Exploring associations between social media addiction, social media fatigue, fear of missing out and sleep quality among university students: a cross-section study 

Dear Dr. Zhu:

I'm pleased to inform you that your manuscript has been deemed suitable for publication in PLOS ONE. Congratulations! Your manuscript is now with our production department. 

Kind regards, 

on behalf of

Dr. Ahsan Ali 

Academic Editor

PLOS ONE